# RNA G-quadruplex structure contributes to cold adaptation in plants

Xiaofei Yang [1,2,3,4,9], Haopeng Yu[1,4,9], Susan Duncan[4], Yueying Zhang[4], Jitender Cheema[4], Haifeng Liu [4,5], J. Benjamin Miller [6], Jie Zhang[4], Chun Kit Kwok [7,8], Huakun Zhang [1] ✉ & Yiliang Ding [4] ✉

Nucleotide composition is suggested to infer gene functionality and ecological adaptation of species to distinct environments. However, the underlying biological function of nucleotide composition dictating environmental adaptations is largely unknown. Here, we systematically analyze the nucleotide composition of transcriptomes across 1000 plants (1KP) and their corresponding habitats. Intriguingly, we find that plants growing in cold climates have guanine (G)-enriched transcriptomes, which are prone to forming RNA G-quadruplex structures. Both immunofluorescence detection and in vivo structure profiling reveal that RNA G-quadruplex formation in plants is globally enhanced in response to cold. Cold-responsive RNA G-quadruplexes strongly enhanced mRNA stability, rather than affecting translation. Disruption of individual RNA G-quadruplex promotes mRNA decay in the cold, leading to impaired plant cold response. Therefore, we propose that plants adopted RNA G-quadruplex structure as a molecular signature to facilitate their adaptation to the cold during evolution.

The earth's prodigious biodiversity was established in part by adaptation to diverse ecological habitats, driving speciation[1]. In particular, plants have become highly evolved to their specific environments, partly due to their sessile nature, with diversity across environments enabling their global colonization[2,3]. Varied nucleotide compositions were suggested to affect plant adaptation to specific habitats[4,5]. However, an understanding of the underlying molecular mechanisms remains unclear.

## Results

### Transcriptomic guanine (G) and RNA G-quadruplex frequencies exhibit climatic signatures of plant adaptation

The recent generation of transcriptome sequences of over 1000 plants (The 1000 plants initiative, 1KP) has allowed us to systematically assess the biological significance of nucleotide compositions across the plant kingdom[6]. We analyzed the frequency of the four nucleotides (A: adenine, U: uracil, C: cytosine, and G: guanine) across all major clades of land plants, including dicots, monocots, gymnosperms, ferns, lycophytes, and bryophytes (Fig. 1a and Supplementary Data 1). Overall, the frequencies of both G and C were consistently lower than those of A and U (Fig. 1a). Frequencies among the four nucleotides varied much more in the 3'-UTR compared to 5'-UTR and CDS (Fig. 1a). We then obtained the habitat locations of these plants from the Global Biodiversity Information Facility (GBIF) and extracted the corresponding 19 associated bioclimatic variables for these habitats from the WorldClim database (Fig. 1b and Supplementary Data 2, see Methods)[7]. We calculated the Pearson Correlation Coefficient (PCC) between the

[1]Key Laboratory of Molecular Epigenetics of Ministry of Education, Northeast Normal University, Changchun 130024, China. [2]National Key Laboratory of Plant Molecular Genetics, CAS Center for Excellence in Molecular Plant Sciences, Institute of Plant Physiology and Ecology, Chinese Academy of Sciences, Shanghai 200032, China. [3]CAS-JIC Center of Excellence for Plant and Microbial Sciences, Institute of Plant Physiology and Ecology, Chinese Academy of Sciences, Shanghai 200032, China. [4]Department of Cell and Developmental Biology, John Innes Centre, Norwich Research Park, Norwich NR4 7UH, United Kingdom. [5]State Key Laboratory of Crop Biology, College of Agronomy, Shandong Agricultural University, Taian 271018, China. [6]School of Biological Sciences, University of East Anglia, Norwich Research Park, Norwich NR4 7TJ, United Kingdom. [7]Department of Chemistry and State Key Laboratory of Marine Pollution, City University of Hong Kong, Kowloon Tong, Hong Kong SAR, China. [8]Shenzhen Research Institute of City University of Hong Kong, Shenzhen 518057, China. [9]These authors contributed equally: Xiaofei Yang, Haopeng Yu. ✉e-mail: zhanghk045@nenu.edu.cn; yiliang.ding@jic.ac.uk

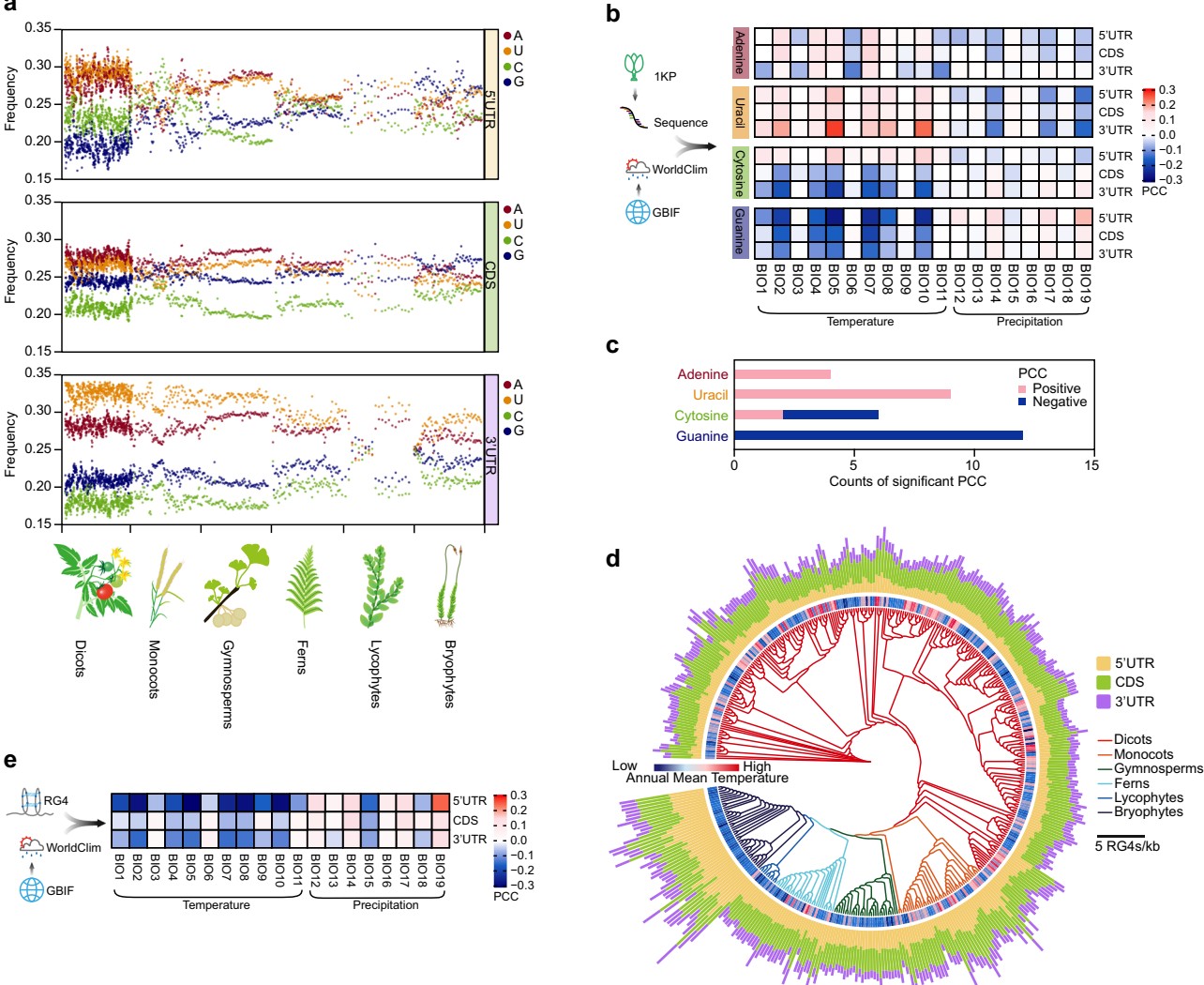

**Fig. 1 | Comparison of transcriptomic guanine (G) frequencies and RNA G-quadruplex frequencies exhibit climatic signatures of plant adaptation.**
**a** Scatter plot showing the frequency of each nucleotide (A: adenine, U: uracil, C: cytosine, and G: guanine) in transcriptomes of 906 land plants from the 1000 Plants (1KP Initiative)[6]. *n* = 556, 107, 80, 71, 21, 71 for dicots, monocots, gymnosperms, ferns, lycophytes and bryophytes, respectively. **b** Heat plot showing the Pearson Correlation Coefficient (PCC) between transcriptomic nucleotide frequency and bioclimatic variables of plant habitats. The frequencies of the four nucleotides were calculated in 906 land plants derived from the transcriptomes of the 1000 Plants (1KP). The corresponding plant habitats were sourced from the Global Biodiversity Information Facility (GBIF). The bioclimatic variables of these plant habitats were derived from the WorldClim database, i.e., BIO1: annual mean temperature (for full details see Materials and Methods), as explained by WorldClim (https://www.

worldclim.org/). **c** Bar plot showing the counts of significant PCCs between transcriptomic nucleotide frequency and temperature bioclimatic variables. PCCs with a *P* value less than a threshold of 0.01 were retained. **d** RNA G-quadruplex (RG4) frequencies in major clades of land plants. Land plants with over 100 occurrences in Global Biodiversity Information Facility (GBIF)[7] were included (see Methods). *n* = 277, 43, 30, 30, 10, 43 for dicots, monocots, gymnosperms, ferns, lycophytes and bryophytes, respectively. The corresponding annual mean temperatures for habitats of plant species were color-coded. **e** Heat plot showing the Pearson Correlation Coefficients (PCCs) between RG4 frequency in plant transcriptomes and associated bioclimatic variables related to plant habitats. PCCs with a *P* value less than a threshold of 0.01 were retained. Plant habitats were sourced from the GBIF, while associated bioclimatic variables were derived from the WorldClim database (full details see Methods).

transcriptomic nucleotide frequencies and bioclimatic variables primarily concerned with temperature and precipitation (i.e., BIO1: annual mean temperature, BIO12: annual precipitation; full list in Methods). PCCs between nucleotide frequency and temperature bioclimatic variables (BIO1-BIO11) were overall stronger than those between nucleotide frequency and precipitation bioclimatic variables (BIO12-BIO19, Fig. 1b), suggesting a stronger relationship between transcriptomic nucleotide composition and temperature of plant habitats. A and U frequencies are mostly positively correlated with temperature bioclimatic variables, while G and C compositions are mostly negatively correlated (Fig. 1c). The profound negative correlation between G frequency and temperature bioclimatic variables was notable, indicating enrichment of G in species growing in cold climates (Fig. 1c).

Given that a G-rich region in an RNA molecule is capable of folding into a tertiary RNA structure called RNA G-quadruplex (RG4), involving the base pairs on both Hoogsteen and Watson-Crick faces (Supplementary Fig. 1a)[8], we, therefore, hypothesized that G-enriched transcriptomes from plants prevalent in colder climates may be complemented with RG4 motifs. We then calculated the RG4 frequency across the 1KP dataset, yielding a median frequency value of 1.27, 2.35, and 0.99 RG4s/kb of RG4 for 5′-UTR, CDS, and 3′-UTR, respectively (Fig. 1d and Supplementary Data 3). Across all genic regions, the overall correlations between RG4 frequency and temperature bioclimatic variables were significantly negative, whilst generally less significant correlations between RG4 frequency and precipitation bioclimatic variables were observed (Figs. 1d, e),

suggesting that RG4 is more strongly enriched in plant species growing in colder climates.

## The folding state of RG4s is globally enhanced in response to cold

To understand the underlying mechanisms of higher RG4 enrichment in plant adaptation to cold, we first visualized RG4 in cells from the model plant *Arabidopsis thaliana*, using the BG4 antibody, which is able to detect RNA G-quadruplex in cells[9]. We found a significant increase and enhancement of cytosolic BG4 foci in the cold at 4 °C, compared to the normal temperature control at 22 °C (Fig. 2a, $P = 10^{-4}$, by Student's *t* test). The increase of the BG4 foci in the cold mimicked RG4 stabilization using the well-characterized pyridostatin ligand (PDS) (Fig. 2a, $P = 10^{-6}$ by Student's *t* test)[10], indicating that cold promotes RG4 folding in plant cells. Notably, when cold-treated plants were returned to 22 °C for 2 hours, BG4 signals recovered to a similar level to that before cold treatment (Fig. 2a, $P = 0.17$ by Student's *t* test).

We then determined the in vivo folding status of individual RG4 motifs in the cold using SHALiPE-seq, a high throughput method for RG4 detection[11]. SHALiPE-seq is based on the preferable modification of the last G residues of G-tracts in the folded RG4 by 2-methylnicotinic acid imidazolide (NAI) (Supplementary Fig. 1b)[11–14] and therefore detects folded RG4 directly. *Arabidopsis* seedlings grown at 22 °C were treated for 2 h either at 4 °C or 22 °C (control) followed by treatment with NAI at 4 °C or 22 °C, respectively, before generating SHALiPE-seq libraries for both groups (Supplementary Fig. 1b). Comparable NAI modification efficiencies under different temperatures were achieved

(Supplementary Figs. 2, 3), by both gel-based analysis and SHALiPE-seq library analysis on the 18 S rRNA.

Reads distribution for Gs of SHALiPE-seq was characterized using the Gini index, where a high Gini index indicates an uneven distribution of reads on Gs for folded RG4s, whilst a low Gini index indicates an even distribution of unfolded RG4s[11,14]. We initially established Gini index benchmarks for folded or unfolded RG4s under in vitro conditions with either $K^+$ (stabilizes RG4) or $Li^+$ (destabilizes RG4) (Supplementary Fig. 1b)[11]. We then implemented a quantitative algorithm to calculate the folding status in vivo, which we termed 'folding score': a comparison of Gini in vivo with Gini in vitro in the presence of $Li^+$, scaled relative to Gini in vitro with $K^+$ vs Gini in vitro with $Li^+$ (see methods)[11]. The folding scores at 4 °C were overall significantly higher compared to those at 22 °C (Fig. 2b, $P < 10^{-148}$, by paired Student's *t* test, Supplementary Data 4), further supporting that stronger RG4 folding occurs in the cold. In particular, some RG4s showed a switch from unfolded to folded status as the temperature shifted from 22 °C to 4 °C. In vivo SHALiPE-seq profile examples of RG4s on *AT3G20470* at 22 °C resembled that of the unfolded in vitro benchmark in the presence of $Li^+$, while corresponding in vivo SHALiPE-seq profiles at 4 °C resembled that of the folded in vitro benchmark in the presence of $K^+$ (Fig. 2c). We further performed circular dichroism (CD) spectroscopy on the RG4 on *AT3G20470* with both $K^+$ titration and temperature shifts. We found that the folding status of this RG4 was enhanced from 22 °C to 4 °C (Supplementary Fig. 4), further supporting our observations in vivo. Therefore, our SHALiPE-seq data along with our immunofluorescence visualizations and biophysical characterizations

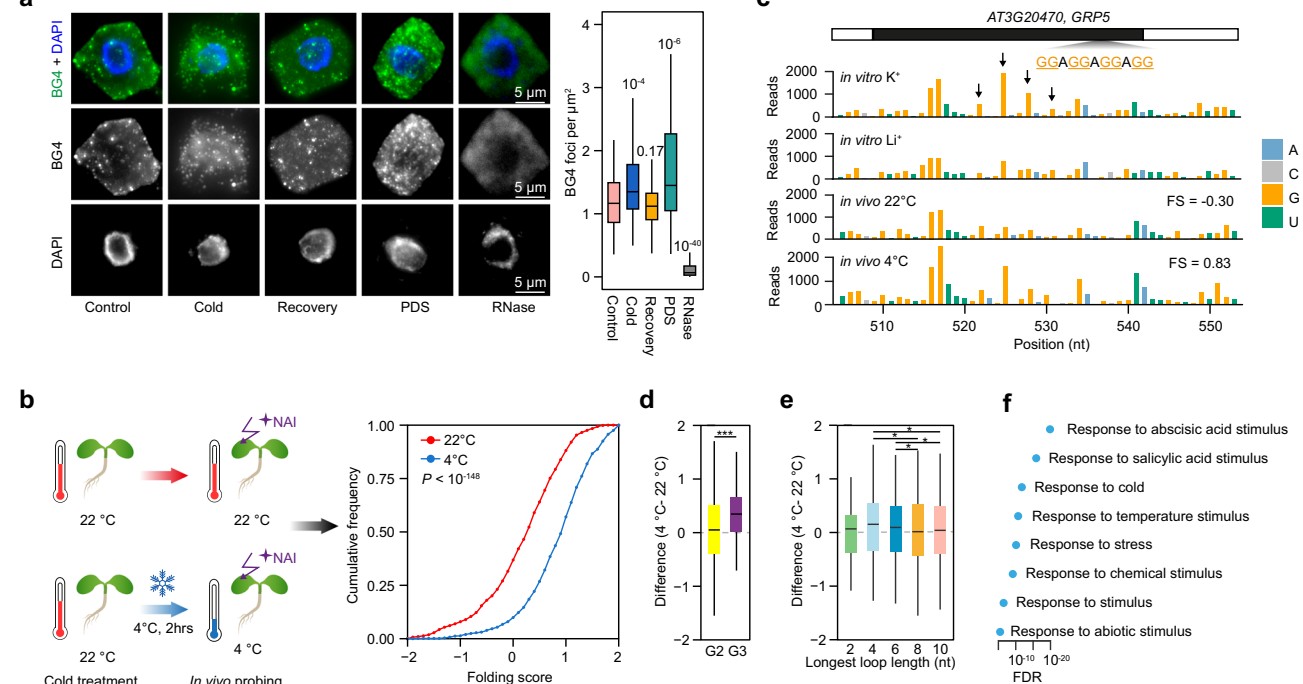

**Fig. 2 | Cold promotes RNA G-quadruplex folding in *Arabidopsis*.**
**a** Immunofluorescence detection of G-quadruplex (RG4) with BG4 antibody in *Arabidopsis* under different thermal conditions. *Arabidopsis* seedlings were grown at 22 °C (Control) and were (i) treated at 4 °C for 2 h (Cold); or (ii) returned to 22 °C for 2 h after cold treatment (Recovery); or (iii) treated with 10 μM pyridostatin (PDS); or (iv) treated with RNase cocktail (RNase). DAPI indicates the nucleus signal. Statistical analysis was performed using >80 cells from three individual seedlings for each treatment, with significance tested by Student's *t* test in comparison to Control. **b** Comparison of RG4 folding by SHALiPE-seq in plants grown at 22 °C and treated either at 4 °C or 22 °C, respectively, $n = 817$, $P < 10^{-148}$, by one-sided Student's *t* test. SHALiPE-seq data were generated from two independent biological

replicates. **c** SHALiPE-seq profiles of the RG4s on *AT3G20470* (*GRP5*). High reads on the last Gs (dark arrow) are compared between in vitro benchmarks: either with $K^+$ (RG4 folded) or $Li^+$ (RG4 unfolded). In vivo profiles show an unfolded state at 22 °C (resembles the benchmark with $Li^+$) but folded state at 4 °C (resembles that with $K^+$). **d** Frequency of distinct RG4 motifs comparing folding score differences (4−22 °C) between G2-RG4 and G3-RG4 motifs; $n = 1436$, 51 for G2 and G3 sites, respectively, ***$P = 0.006$ by one-sided Student's *t* test. **e** Comparison of RG4 folding score difference frequencies (4−22 °C) with different RG4 loop lengths. RG4 sites $n = 104$, 304, 243, 442, and 343 for loop length of 2nt, 4nt, 6nt, 8nt and 10nt, respectively. *$P < 0.05$ by one-sided Student's *t* test. **f** Representative gene ontology items of transcripts with cold-responsive RG4s.

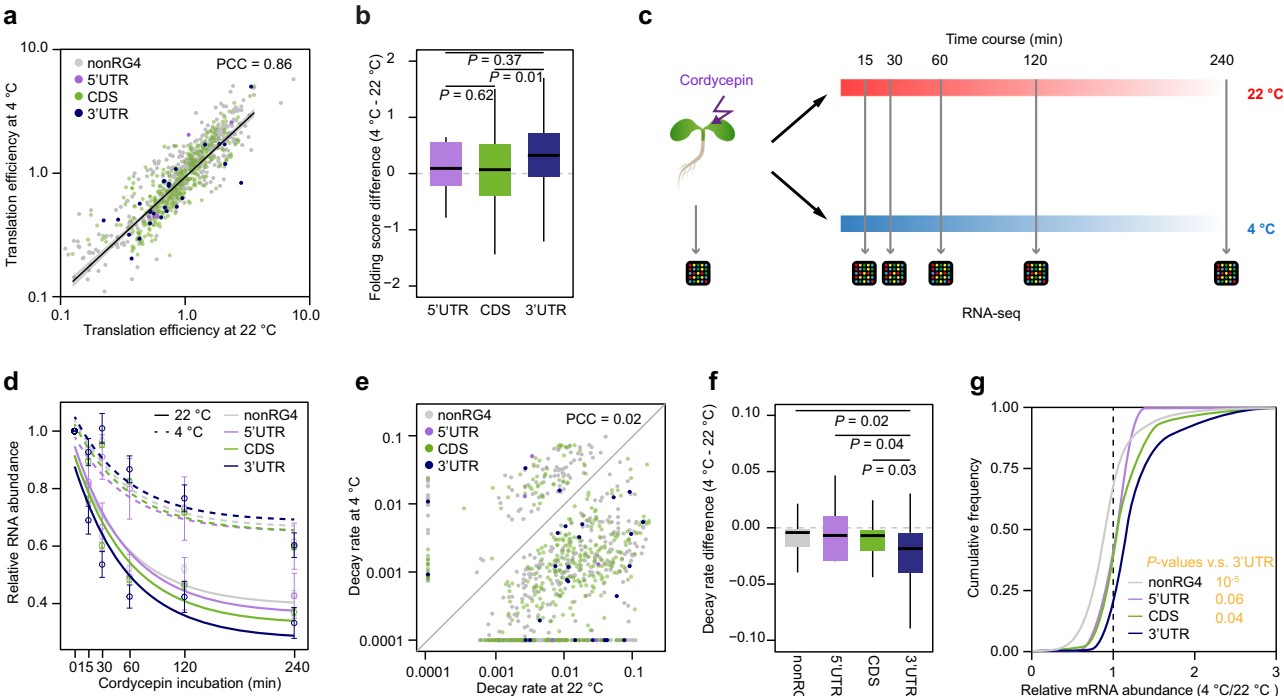

**Fig. 3 | Cold-responsive RNA G-quadruplex in 3′-UTR represses mRNA decay.** **a** Comparison of translation efficiency (TE) at 22 °C and 4 °C, for transcripts without RG4 (nonRG4), with RG4 in 5′-UTR, with RG4 in CDS, and with RG4 in 3′-UTR. The number of transcripts, $n = 465, 6, 531, 36$, for nonRG4, 5′-UTR, CDS and 3′-UTR respectively. TE was calculated based on three independent biological replicates. **b** Comparison of folding score differences (4–22 °C) between RG4 located in different genic regions. The number of RG4 sites, $n = 9, 1421, 57$ for 5′-UTR, CDS, and 3′-UTR, respectively; $P$ values by two-sided Student's $t$ test. **c** Schematic of determining RNA stability in *Arabidopsis* at 22 °C and 4 °C, by transcriptional arrest analysis with cordycepin incubation. **d** RNA abundance curves at 22 °C and 4 °C for transcripts with cold-responsive RG4s, otherwise in **a**. Data are presented as mean values ±SE of different transcripts. RNA abundance was calculated based on three independent biological replicates. **e** Comparison of the decay rate of transcripts with cold-responsive RG4s at 22 °C and 4 °C, otherwise in **a**. **f** Comparison of decay rate differences (4–22 °C) on transcripts with cold-responsive RG4s. $P$ value by two-sided Student's $t$ test, otherwise in **a**. **g** Line plot showing relative mRNA abundance at 4 °C versus 22 °C for transcripts with cold-responsive RG4s. $P$ values by Student's $t$ test, otherwise in **a**.

reinforce the notion that RG4 folding is responsive to cold in living plants.

RG4s have a range of G-quartets and loop lengths[8,15]. To test whether different types of RG4s may be favored in the cold, we compared the folding score differences between 4 °C and 22 °C on individual RG4s. We found greater folding score differences for RG4s with three G-quartets (G3-RG4) compared to those with two G-quartets (G2-RG4, Fig. 2d, $P = 0.006$, by Student's $t$ test). In addition, the folding status of RG4s with intermediate lengths from 4nt to 6nt was more easily enhanced in the cold, compared to RG4s with either a very short loop of 2nt or a very long loop of over 8nt (Fig. 2e, $P < 0.05$, by Student's $t$ test). Thus, cold conditions are likely to strongly promote the folding of RG4s with a higher number of G-quartets and intermediate loop lengths.

To understand any biological relevance between RG4s and the cold, we performed gene ontology (GO) analysis on genes containing cold-responsive RG4s with higher folding scores upon cold treatment. We found specific transcript-related enrichment in biological functions, such as response to abiotic stimulus, response to temperature stimulus, and response to cold (Fig. 2f). Thus, those genes containing cold-responsive RG4s are also likely to function in plant response to cold.

### The cold-responsive RG4s in the 3′-UTR serve as mRNA stabilizers

To further explore how these cold-responsive RG4s may affect the plant response to cold, we assessed their molecular functions in regulating gene expression. Extensive studies have suggested that RG4 serves as a translational repressor[8,11,16]. Thus, we assessed whether cold-responsive RG4s repress translation in the cold. We compared the translation efficiencies (TEs) of those genes containing cold-responsive RG4s at either 4 °C or 22 °C. Overall, we observed only a subtle variation (-2%) and a high correlation between TE at 4 °C and TE at 22 °C (Fig. 3a, PCC = 0.86, difference = 2%, Supplementary Data 5), suggesting cold-responsive RG4s are unlikely to contribute much to translation. We then examined the folding score differences between 4 °C and 22 °C of RG4s in different genic regions and found RG4s in the 3′-UTR possessed the greatest folding score differences (Fig. 3b, $P = 0.01$ for CDS vs 3′-UTR, by Student's $t$ test). Based on previous studies that suggested structural elements in 3′-UTRs are likely to regulate mRNA stability[17–19], we hypothesized that cold-responsive RG4s may impact mRNA stability. We measured the RNA stability of the *Arabidopsis* transcriptome at both 22 °C and 4 °C using the transcription arrest assay (Fig. 3c)[20]. Upon transcription arrest, the rate of mRNA abundance decline reveals RNA stability, whereby a rapid decrease indicates low stability whilst a slow decrease indicates high stability. Overall, we found a generally slower mRNA abundance decline at 4 °C compared to 22 °C (Fig. 3d). Notably, the cold effect of slowing down RNA abundance decline was found to be highest for transcripts with cold-responsive RG4s in 3′-UTRs, compared to those transcripts without RG4s (nonRG4), or with cold-responsive RG4s in 5′-UTR and/or CDS regions (Fig. 3d). The transcription arrest profiles for individual mRNAs are illustrated for *AT1G13390*, *AT4G32020*, and *AT5G24930* along with their corresponding qRT-PCR validations (Supplementary Fig. 5).

We then derived the mRNA decay rate for individual mRNAs with cold-responsive RG4s at both 4 °C and 22 °C (Supplementary Data 6). The mRNA decay rates at 4 °C altered dramatically compared to those

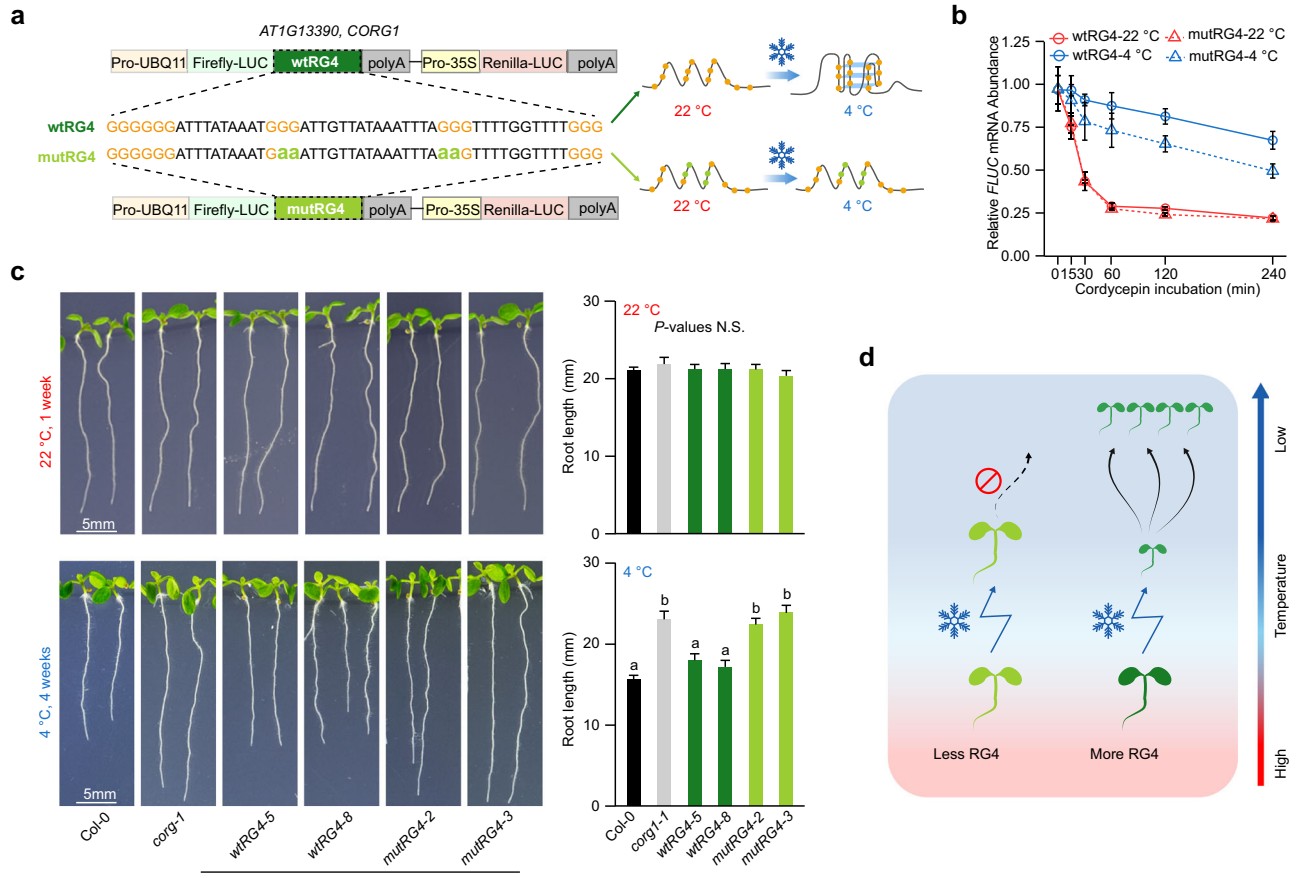

**Fig. 4 | RNA G-quadruplex regulates plant cold-responsive growth. a** Schematic of folding status of wild-type RG4 (wtRG4) and mutated RG4 (mutRG4) on *CORG1* 3′-UTR at 22 °C and 4 °C. The RG4 folding potential was either maintained in wtRG4 or attenuated in mutRG4 by G to A substitution. **b** RNA decay curves at 22 °C and 4 °C for *FIREFLY-LUCIFERASE* (*FLUC*) transcripts with wtRG4 or mutRG4, as with the designs shown in **a**. Data are presented as mean values ± SE, *n* = 4. **c** Phenotypes of plants of different genotypes grown at 22 °C or 4 °C. wtRG4 or mutRG4 denotes *corg1-1* mutant complemented with genomic DNA of *CORG1* carrying wtRG4 or mutRG4 respectively, as illustrated in **a**. Comparison was performed on the plants grown vertically on plates at 22 °C for 1 week, or at 4 °C for 4 weeks. Significance tested by one-way ANOVA/Tukey HSD post hoc test (*P* < 0.05), *n* ≥ 20. Data are presented as mean values ± SE. **d** Model showing the selective enrichment of RG4 driving plant adaptation to cold climates. More RG4s in the plant transcriptome enable higher phenotypic plasticity in cold sensing, therefore contributing to plant adaptation to colder climates.

at 22 °C (Fig. 3d, e, PCC = 0.02), and generally showed slower mRNA decay rates in the cold. We subsequently calculated the decay rate differences between 4 °C and 22 °C and found a significantly greater difference in decay rate between 4 °C and 22 °C on transcripts with cold-responsive RG4s in 3′-UTR compared to that of transcripts without RG4s or with RG4s in 5′-UTR or CDS regions (Fig. 3f, *P* values = 0.02, 0.04 and 0.03 respectively, by Student's *t* test). As mRNA stability directly affects steady-state mRNA abundance, we reasoned that cold enhances the steady-state mRNA abundance of transcripts with cold-responsive RG4s. Indeed, we found a strong increase of mRNA abundance in the cold for transcripts with cold-responsive RG4s in the 3′-UTRs, rather than nonRG4 transcripts or cold-responsive RG4s in 5′-UTR/CDS regions (Fig. 3g, $P = 10^{-5}$, 0.06 and 0.04 respectively, by Student's *t* test). Collectively, our results suggest cold-responsive RG4s strongly impact mRNA stability rather than mRNA translation. These cold-responsive RG4s in the 3′-UTR serve as mRNA stabilizers, reducing mRNA degradation in the cold.

**The cold-responsive RG4 modulates plant cold-responsive growth**

Considering the strong molecular function of these cold-responsive RG4s in 3′-UTRs, we reasoned that these cold-responsive RG4s, as mRNA stabilizers, may affect plant phenotypic response to the cold. Of

the cold-responsive RG4s present in 3′-UTRs, the one with the greatest folding score difference between 4 °C and 22 °C was found on *AT1G13390* (Supplementary Fig. 6a and Supplementary Data 4), designated as *CORG1* (COld-responsive RNA G-quadruplex 1), with the annotation encoding a translocase subunit protein (courtesy of The Arabidopsis Information Resource). To further explore whether this *CORG1* RG4 contributes to mRNA stabilization in the cold, we constructed the reporter gene, *FIREFLY-LUCIFERASE* (*FLUC*) fused with the 3′-UTR of *CORG1* containing the wild-type cold-responsive RG4 (wtRG4) or a disrupted RG4 (mutRG4, Fig. 4a). We then performed transcriptional arrest assays at both 4 °C and 22 °C to measure the corresponding mRNA stabilities. The *FLUC* mRNA with the 3′-UTR of *CORG1* containing the mutRG4 degraded much more rapidly compared to that containing wtRG4 at 4 °C, while no significant differences were found between these two versions at 22 °C (Fig. 4b), indicating that this cold-responsive RG4 enhanced mRNA stabilization in cold conditions.

We then reintroduced *Arabidopsis* genomic DNA of *CORG1* containing 3′-UTR with either wtRG4 or mutRG4 into the *corg1-1* null mutant (Supplementary Fig. 6b, c). At 22 °C, the primary root growth of *corg1-1* was similar to that of wild-type Col-0 (Fig. 4c). At 4 °C, the root length of *corg1-1* was distinctively longer than that of Col-0, indicating *CORG1* represses plant growth under cold conditions. When the *corg1-1* mutant was complemented with *mutRG4-CORG1*, the root

length was significantly longer than *wtRG4-CORG1* complemented plants at 4 °C, while no significant differences were observed at 22 °C (Fig. 4c, *P* values < 0.01, one-way ANOVA/Tukey HSD post hoc test), suggesting impairment of *mutRG4-CORG1* plants in sensing cold. The cold-sensitive defects in the *mutRG4-CORG1* plants correspond to the more rapid RNA decay and decrease in steady-state *CORG1* RNA abundance (Supplementary Fig. 6d, e). Additionally, the general plant sizes of the *corg1-1* mutant complemented with *mutRG4-CORG1* were overall bigger than *wtRG4-CORG1* complemented plants at 4 °C, while no significant differences were observed at 22 °C (Supplementary Fig. 6f). Furthermore, we found that the cold-responsive expression levels of the marker genes for *Arabidopsis* cold response including *CBF1*, *CBF2*, *CBF3,* and *COR15A* were significantly lower in the *corg1-1* mutant complemented with *mutRG4-CORG1* than those in *wtRG4-CORG1* complemented plants upon cold treatments (Supplementary Fig. 7).

## Discussion

Our broad and deep investigations of nucleotide composition across the plant kingdom promote the discovery that RNA G-quadruplex structure may serve as a molecular signature contributing to plant environmental adaptation. Correlations between RG4 frequency and temperature bioclimatic variables are not very high (Fig. 1e), suggesting other factors besides RG4 may also contribute to plant thermal adaptation. RNA G-quadruplexes embedded across the plant transcriptome might play an important role in stabilizing RNA molecules in response to cold (Fig. 4d). Hence, our results suggest that RNA structure, as a key regulator of gene expression[21–23], may have evolved due to selection pressures arising from distinct temperature environments. Our work, therefore, advances our fundamental understanding of the molecular mechanisms underlying species adaptation. Ultimately, the translation of this conceptual advance to molecular engineering strategies could help to address global climate change challenges indicated by continuing increases in the frequency, intensity, and duration of extreme low-temperature events[24,25].

## Methods

### Statistics

No statistical methods were used to predetermine the sample size. The experiments were not randomized, and investigators were not blinded to allocation during experiments and outcome assessment. Sampling in all cases was performed by collecting materials independently from new plants for replicates.

### Plants and growth conditions

The *Arabidopsis thaliana* ecotype Columbia (Col-0) and *corg1-1* mutant (SALK_097604) with T-DNA insertion were obtained from the Nottingham Arabidopsis Stock Centre (NASC); the homozygous mutant was identified using PCR-based genotyping[26]. *Arabidopsis* seeds were sterilized using 70% ethanol for 10 min, washed with distilled water three times, and plated one-half strength Murashige and Skoog medium supplemented with 1% sucrose. After standing at 4 °C for 3 days, the plates were placed in a growth chamber at 22 °C. To complement the mutant, *corg1-1* mutant plants were transformed using a floral dipping method[27]. Transgenic plants were selected using GM media with 10 mg/l phosphinotricin.

### Plasmid construction

To complement the *Arabidopsis* mutants with wild-type sequence, a DNA fragment of *AT1G13390* was amplified from Col-0 genomic DNA using CloneAmp HiFi PCR Premix (Clontech) with primers listed in Supplementary Data 7. To complement the *Arabidopsis* mutants with disrupted RG4 sequence (mutRG4), fragments were amplified by overlap PCR with designed mutations (guanine -> adenine, G -> A) on primers (Supplementary Data 7)[28]. PCR products were further introduced into the XmnI and KpnI digested pB7FWG2.0 with In-Fusion

(Clontech). For dual-luciferase analysis, the sequence of 3′UTR of *AT1G13390* with or without mutation was cloned into the expression vector inter2 digested with AscI and PstI as introduced with designed primers (Supplementary Data 7). Sequencing-confirmed vectors were then transformed to *Agrobacterium tumefaciens* GV3101.

### Immunofluorescence detection of RG4

*Arabidopsis* seedlings of 5 days after germination (DAG) were fixed in 4% methanol-free formaldehyde (Thermo Scientific) diluted in nuclease-free 1× PBS (Invitrogen) for 30 min, then washed three times with 1× PBS (Invitrogen). Roots were then squashed under coverslips onto slides that had been pre-treated with 3-Aminopropylthiethoxysilane (Sigma-Aldrich) to increase tissue adherence[29]. Samples were then immersed in liquid nitrogen for around 10 seconds before the coverslips were removed. The tissue was left to dry for 1 h at room temperature, 1× PBS with 0.1% Triton X-100 was added to each slide, incubated for 3 mins following with three washes with 1× PBS. At this stage RNase treatment was carried out for a subset of +RNase slides by adding 12U RNase A and 200U RNase T1 (Invitrogen) diluted in 1× PBS to each slide and incubated for 1.5 h at 37 °C followed by three 5-min washes with 1× PBS. All slides were immersed in 70% ethanol for 2 h at room temperature. After three 1× PBS washes, 2% BSA dissolved in 1× PBS was added to each slide and left to incubate at room temperature for 1 h. The block was aspirated and anti-DNA/RNA G-quadruplex Mouse Fab fragment antibody (BG4 antibody, His-Tagged, Lambda antibody AB00174-1.6, Absolute Antibody) with a dilution of 1:300 in 1× PBS with 2% BSA was added to each slide and incubated in a dark[9], humid box overnight at 4 °C. Three 5-min washes with 1× PBS were then carried out and 1:1000 diluted Alexa Fluor 647 goat anti-mouse IgG H + L secondary antibody A21235 (Invitrogen) was added to each slide, followed by incubation at room temperature for 1 h. Two 5-min washes with 1× PBS were carried out followed by a 30 min incubation at room temperature with 1 µg/mL DAPI in 1× PBS, followed by three 5-min washes with 1× PBS and three washes with nuclease-free water. The samples were mounted with ProLong Glass (Thermo Fisher Scientific) under No. 1.5 coverslips (VWR). Root cells were imaged using a 100x objective (NA 1.46) on an inverted ZEISS Elyra PS1 microscope (ZEISS, Oberkochen) equipped with an Andor EM-CCD camera and controlled by ZEN 2.3 SP1 software (ZEISS, Oberkochen). 0.2 µm optical Z-sections were carried out and images were obtained using 405 nm and 642 nm laser lines. ImageJ plugin RS-FISH was used for automated G-quadruplex quantification as described[30,31]. Analysis was carried out on 2D maximum intensity Z-projections for individual cells with default settings apart from optimization of difference-of-gaussian values and intensity threshold. To perform PDS treatment in *Arabidopsis*, seedlings were incubated with 10 µM PDS dissolved in half strength of Murashige and Skoog liquid medium for 2 h. PDS penetration was carried out by vacuum infiltration–three times for 30 seconds and 1 min break each. To perform cold treatment on *Arabidopsis*, the plate was placed into a growth chamber MIR-254-PE (Panasonic) at 4 °C for 2 h.

### In vivo NAI probing of RNA

The NAI probing buffer was kept in an incubator at the desired temperature overnight before use. NAI probing at 22 °C was carried out as described in our previous study[11]. For NAI probing at 4 °C, plates with 5-DAG etiolated seedlings were treated at 4 °C for 2 h. Treated seedlings were harvested in a cold room and incubated with 400 mM NAI at 4 °C for 15 min. NAI was quenched using five times DTT to that of NAI. Seedlings were washed three times with distilled water, ground in liquid nitrogen, and applied to RNA extraction using RNeasy Plant Mini Kit (Qiagen).

### Gel-based analysis of NAI probing

RNA was extracted after NAI probing. 1 µg of total RNA was dissolved in 6 µL water, 1 µL of 5 µM Cy5 modified RT primer for 18 S rRNA (listed in

Supplementary Data 7) and 0.5 µL of 10 mM dNTPs were denatured at 95 °C for 3 min. The reaction was cooled down to 50 °C before adding 2 µL of 5× home-made RT buffer (100 mM Tris (pH 8.3), 500 mM LiCl, 15 mM MgCl$_2$, 5 mM DTT) and 0.5 µL reverse transcriptase Superscript III (Invitrogen) and mixing quickly with a pipette. The RT reaction was incubated at 50 °C for 20 min followed a staying at 85 °C for 10 min to inactivate reverse transcriptase. cDNA hybridized RNA was degraded by adding 0.5 µL of 2 M NaOH and incubating at 95 °C for 10 min. Equal volumes of 2× stopping dye (95% formaldehyde, 20 mM EDTA (pH 8.0), 20 mM Tris (pH 7.5), orange G) were added and incubated at 95 °C for 5 min. The resultant reaction was kept at 65 °C and loaded to 8% Acrylamide: Bis-Acrylamide-Urea gel for electrophoresis. For the sequencing lanes, RNA was dissolved in 5 µL water, and 1 µL 10 mM of corresponding ddNTP (Roche) was added at the beginning.

### Generation of the SHALiPE-seq libraries

Libraries of SHALiPE-seq were prepared as described[11,32]. PolyA-selected RNA was recovered and reverse transcribed using super-script III (Invitrogen) and RT primer (5′CAGACGTGTGCTCTTCCG ATCTNNNNNNN3′) with home-made RT buffer (20 mM Tris (pH 8.3), 100 mM LiCl, 3 mM MgCl$_2$, 1 mM DTT). The 3′-end of resultant cDNAs were ligated to an ssDNA linker (5′-PhosNNNAGATCGGAAGAGC GTCGTGTAG-/3SpC3/3′) using Circligase ssDNA Ligase (Epicentre) at 65 °C for 12 hrs. Product longer than 100 nt was recovered using QIAquick Gel Extraction Kit (Qiagen) after separation of the ligation products with TBE-Urea 10% Gel (Invitrogen). Purified cDNA was subjected to PCR amplification using KAPA Library Amplification Kits (Roche) with Forward Library primer (5′AATGATACGGCGACCA CCGAGATCTACACTCTTTCCCTACACGACGCTCTTCCGATCT3′) and Reverse Library primers (5′CAAGCAGAAGACGGCATACGAGATNNNNN NGTGACTGGAGTTCAGACGTGTGCTCTTCCGATC3′, where NNNNNN denotes the barcodes, e.g. Index1 is CGTGAT for Illumina sequencing). Three rounds of agarose gel purification were performed to purify the fragments of 200-650 bp using QIAquick Gel Extraction Kit (Qiagen). The libraries were sequenced with Illumina HiSeq 4000 platform by BGI Genomics (Hong kong).

### Polysome-associated mRNA analysis and RNA-seq

Polysome-associated mRNA analysis was performed as described with modifications[33,34]. Briefly, ~500 mg 5-DAG etiolated seedlings (grown at 22 °C, with or without 2 h treatment at 4 °C) were harvested and ground into fine powder in liquid nitrogen. The powder was dissolved in 500 µL pre-cooled polysome extraction buffer (200 mM Tris-HCl, pH 8.4, 50 mM KCl, 1% deoxycholic acid, 25 mM MgCl$_2$, 2% Poly-oxyethylene 10 tridecyl ether, 2 mM DTT, 400 U/mL recombinant Rnasin, 50 µg/mL cycloheximide). After incubation on ice for 30 min and centrifugation at $16{,}000 \times g$ for 15 min at 4 °C, 500 µL supernatant was transferred to 15–60% sucrose gradient. Polysomes were fragmented by 4 h of centrifugation at $28{,}000 \times g$ in Beckman ST40Ti rotor. A low to high sucrose gradient was collected, taking fractions 1–12. Fractions of 7–12 representing the translation level were taken for RNA isolation with TRIzol reagent (Ambion). To perform an RNA-seq experiment, around 25 mg plant powder was used for direct RNA isolation by RNeasy Plant Mini Kit (Qiagen). Polysome-bound RNA or total RNA was subjected to RNA-seq library generation by BGI Genomics following the manufacturer's BGISEQ-500 protocol.

### Transcriptional arrest

Transcriptional arrest analysis was performed as described with modifications[20]. For analysis in *Arabidopsis*, seedlings were harvested and placed into a petri dish with two layers of filter paper and incubation buffer with cordycepin (1 mM PIPES, pH 6.25, 15 mM sucrose, 1 mM potassium chloride, 1 mM sodium citrate, and 1 mM cordycepin). For reporter assay in tobacco (*N. benthamiana*) leaves after 48 h of agroinfiltration were harvested and cut into small discs (~5 mm × 5

mm). To promote the penetration of cordycepin, vacuum infiltration was performed 3 times, for 1 min each. The *Arabidopsis* seedlings or tobacco leaf discs incubated with cordycepin were placed either at 22 °C or 4 °C and harvested using a time series of 15, 30, 60, 120, and 240 min. For cold treatment experiments, the buffer was pre-cooled at 4 °C, vacuum infiltration was performed in a cold room at 4 °C. The plants without cordycepin incubation were harvested as the control, corresponding to 0 min of cordycepin incubation. RNA was extracted and subjected to either quantitative real-time PCR (qRT-PCR) or RNA sequencing.

### Quantitative real-time PCR

RNA was digested using RNase-free TURBO™ DNase (Ambion). First-strand cDNA was synthesized using reverse transcriptase Superscript III (Invitrogen) and oligo dT primer with home-made RT buffer (20 mM Tris (pH 8.3), 100 mM LiCl, 3 mM MgCl$_2$, 1 mM DTT). Quantitative qRT-PCR was performed with LightCycler® 480 SYBR Green I Master (Roche) using CFX96 Touch Real-Time PCR Detection System (BIORAD) according to the manufacturer's protocol. PP2A (AT1G13320) was used as the internal control. All primers are listed in Supplementary Data 7.

### Phenotype assessment

To measure primary root length, images of 1-week seedlings grown at 22 °C under short-day (SD, 8/16 h, light/dark) were captured using a digital camera. The root length of more than 20 seedlings was measured using ImageJ software (NIH). For cold treatment at 4 °C, plates were placed at 22 °C for 2 days for germination and transferred to 4 °C for further growth under the same light condition. Images were taken after 4 weeks' growth at 4 °C.

### RNA G-quadruplex structure (RG4) prediction

Plant transcriptomes were obtained from the resource data of the one thousand plant transcriptomes project[6]. RNA G-quadruplex features were extracted according to our previous study[11], and further assigned to different structural subclasses of G2-RG4 (RG4 with two layers of G-quartet) or G3-RG4 (RG4 with three layers of G-quartet). The RG4 density was calculated by the count of predicted RG4s, normalized to the total amount of all four bases in the transcriptome for individual species.

### Bioclimatic variables for plants

The latitude and longitude of plant geographic distribution were obtained from observations collected by the Global Biodiversity Information Facility (www.gbif.org, i.e. Arabidopsis thaliana: 10.15468/dl.pmy46j, full list see Supplementary Data 2). The 433 species with over 100 observations were included in further analysis. The nineteen bioclimatic variables at representative locations were extracted from the WorldClim database[7]. BIO1:annual mean temperature; BIO2: mean diurnal range (mean of monthly (max temp−min temp)); BIO3: iso-thermality (BIO2/BIO7) (×100); BIO4: temperature seasonality (standard deviation ×100); BIO5: max temperature of the warmest month; BIO6: min temperature of the coldest month; BIO7: temperature annual range (BIO5-BIO6); BIO8: mean temperature of the wettest quarter; BIO9: mean temperature of the driest quarter; BIO10: mean temperature of the warmest quarter; BIO11: mean temperature of the coldest quarter; BIO12: annual precipitation; BIO13: precipitation of wettest month; BIO14: precipitation of driest month; BIO15: precipitation seasonality (coefficient of variation); BIO16: precipitation of wettest quarter; BIO17: precipitation of driest quarter; BIO18: precipitation of warmest quarter; BIO19: precipitation of coldest quarter. We obtained the corresponding climate features from the latitude and longitude of each plant observation and calculated the 10, 25, 50, 75, and 90 quartiles. The Pearson Coefficient Correlation analysis was performed by R scripts, and the corresponding *P* values were adjusted

by the Benjamini–Hochberg method (FDR). The lowest FDR among the different quartiles of climate features was used to represent the corresponding features[4].

## Alignment of the sequencing reads

For RNA-seq and polysome-seq libraries, reads were directly used for alignment. For SHALiPE-seq libraries, the first 3 bases at the 5′-end (random nucleotides on adaptor for cDNA ligation) of the raw reads were cropped. The alignment was carried out against *Arabidopsis* transcriptome TAIR10 release using bowtie version 1.0.1 with iterative mapping procedure[35]. The minimum read length allowed to map was fixed to 21 bases long[36]. The resulting mapped sam files were converted to bam files and indexed using samtools-1.4.1[37]. The stop counts were extracted using HTseq v0.7.2 and the code was written in Python v2.7.15. Reads counts of replicates were merged after observing high correlations among these.

## Gini index and folding score calculation

Gin index was calculated from the SHALiPE-seq libraries with reads number of G residues in G-tract as described[11,14].

$$\text{Gini} = \frac{\sum_{i=1}^{n}\sum_{j=1}^{n}|ri - rj|}{2n^2\bar{r}} \tag{1}$$

$n$ denotes the number of *G* residues in the G-tracts, and $ri$ denotes the reads number in SHALiPE profiling at position $i$.

To calculate the folding score, the regions with Gini (in vitro K$^+$)/ Gini (in vitro Li$^+$) ≥1.1 and *G* residues with average reads count ≥10 were included.

$$\text{folding score} = \frac{\text{Gini(in vivo)} - \text{Gini(in vitro Li}^+)}{\text{Gini(in vitro K}^+) - \text{Gini(in vitro Li}^+)} \tag{2}$$

## GO analysis

Enrichment analysis of GO categories was performed online by AgriGo[38].

## Reporting summary

Further information on research design is available in the Nature Research Reporting Summary linked to this article.

## Data availability

The raw sequencing data have been deposited in the Sequence Read Archive (SRA) (https://www.ncbi.nlm.nih.gov/sra) under BioProject ID number PRJNA762705. SHALiPE-seq data in vitro and in vivo at 22 °C are available under BioProject ID number PRJNA561194[11]. The processed data are provided in the Supplementary Data 1–7. Resources for the latitude and longitude of plant geographic distribution obtained from observations collected by the Global Biodiversity Information Facility (www.gbif.org) have been listed in Supplementary Data 2, i.e. *Arabidopsis thaliana*: https://doi.org/10.15468/dl.pmy46j. WorldClim database could be accessed through the server https://worldclim.org/.

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

## Acknowledgements

The study is supported by the National Natural Science Foundation of China (32170229) (H.Z.), the National Key Research and Development Program of China (2021YFF1000900) (H.Z.), United Kingdom Biotechnology and Biological Sciences Research Council (BBSRC: BBS/E/J/000PR9788, BB/L025000/1, and BB/N022572/1) (Y.D.), Norwich Research Park Science links seed fund (Y.D.), the European Research Council (ERC: 680324) (Y.D.), Human Frontier Science Program Fellowship (LT001077/2021-L) (H.Y.), Shenzhen Basic Research Project (JCYJ20180507181642811) (C.K.K.), Research Grants Council of the Hong Kong SAR, China Projects (CityU 11100421, CityU 11101519, CityU 11100218 and N_CityU110/17) (C.K.K.), Croucher Foundation Project (9509003)(C.K.K), State Key Laboratory of Marine Pollution Director Discretionary Fund (C.K.K.). We thank professor Beverley Glover (U. Cambridge), Dr. Sam Brockington (U. Cambridge), professor Shiheng Tao (Northwest A&F University), professor Giles Oldroyd (SLCU, Cambridge), and Dame professor Caroline Dean (John Innes Centre), Dr. Antony Dodd (John Innes Centre), professor Cristobal Uauy (John Innes Centre), professor Alison Smith (John Innes Centre) and Dr. Desmond Bradley (John Innes Centre) for discussions with this work. We thank the John Innes Centre Bioimaging facility and staff for their contribution to this publication. This research was supported by the Norwich Bioscience Institutes Partnership's Computing infrastructure for Science (CiS) group through the provision of a High-Performance Computing Cluster and the John Innes Centre Informatics team.

## Author contributions

Y.D. conceived the study; X.Y., H.Y., J.B.M., C.K.K., H.Z., and Y.D. designed the study; X.Y., S.D., Y.Z., H. L., and J.Z. performed the experiments; H.Y., X.Y., S.D., and J.C. did the analyses; H.Z. and Y.D. supervised the analyses; X.Y., H.Y., S.D., J.C., H.Z., and Y.D. wrote the paper with input from all authors.

## Competing interests

The authors declare no competing interests.

## Additional information

**Correspondence and requests** for materials should be addressed to Huakun Zhang or Yiliang Ding.

