## [Peer Review File · Nature Communications]

Title: RNA G-quadruplex structure contributes to cold adaptation in plantsREVIEWER COMMENTS

Reviewer #1 (Remarks to the Author):

In their manuscript Yang et al. investigate the role of rG4 in plant transcript as a potential marker for cold response. The authors combine a series of bioinformatic analysis with rG4 mapping strategies (SHALIPE) to address the potential biological role of rG4 structure in mRNA stability and plant growth. Using BG4-staining counting cytoplasmic foci (as described in seminal paper by Biffi et al Nat Chem 2013) they quantify a global increase of rG4 formation in cold condition, which is supported by the use of a positive control ligand PDS. By using transcriptome-wide approaches the authors then demonstrate that transcript with rG4 increases mRNA stability. This experiment is particularly elegant and convincing as it is performed on a transcriptome-wide scale and not only of few selected transcript. Furthermore, the authors demonstrate how the presence of such rG4 can be relevant for plant development. Overall, this is a very strong and elegant paper with robust evidence to support the authors conclusion. Therefore I am supportive of publication in Nat Comm.

I have only some minor comments for the authors to consider prior publication:

- 1) The nature of rG4 involved in the mechanism described seems to be multi-tetrad long looped (as mentioned by the authors). It would be interesting to have a UV or CD melting curve for some of these sequences to confirm that the switching temperature between folded/unfolded rG4 is somewhere between 4 and 22 C.
- 2) On a similar note it would be good to have some biophysical characterisation of some of the rG4 sequences identified (i.e. CD spectra), just to confirm that those sequences are indeed folding into a G4, although the SHALIPE data are already pretty convincing, but just for completeness.
- 3) I found extended data Fig.1 extremely compelling, I wonder whether there is scope for this to be in the main text

Reviewer #2 (Remarks to the Author):

Yang et al analyzed the nucleotide compositions of transcriptomes across plants and conclude that those living in cold climates have G-quadruplex enriched transcriptomes. One thing is to observe that different species living at high or low temperatures have different rG4 propensities. Another entirely different topic is to study G4 formation in a given plant grown at different temperatures. The latter seems relatively trivial: RNA secondary structures, including G-quadruplexes, should be more stable at low temperature. One would expect a similar behavior in human cells for example. Has this been tested ?

Fig 1a: The fact that both G and C are enriched in cold-living plants is not necessarily a proof that G-

quadruplexes are actually involved (one would expect only the fraction of G to increase – what is seen here is more like an increase in GC content)

Fig 1b: the warm/cold data is strongly skewed by bryophytes: all of them live in cold environments, and their G4 density is clearly higher than in other plant species.

Some of the final statements need to be tuned-down, such as “G-quadruplex structure serving as a molecular marker shaping plant environmental adaptation” or “RNA G-quadruplexes embedded across the plant transcriptome globally facilitate stabilizing RNA molecules in response to cold”.

Detail: “RNA G-quadruplex (RG4), involving both Hoogsteen and Watson-Crick base pairs” is a bit misleading – most quadruplexes involved only Hoogsteen type base pairing.

Overall, while potentially interesting, this manuscript is more appropriate for a specialized plant journal.

Reviewer #3 (Remarks to the Author):

The authors analysed nucleotide compositions of transcriptomes across 1000 plants and compared them with their corresponding climate to observe an interesting correlation between G-rich transcriptomes and G-quadruplex structures and cold acclimated plants. They use immunofluorescence and a variety of genomic approaches (including SHALiPEseq, cordycepin treatments for mRNA seq) to assess the impact of G4 quartets on translation efficiency and mRNA stability at different temperatures and find a strong correlation with cold responses. Then, they identified a specific target CORG1, a cold regulated G4-quartet responsive transcript, to address the relevance of the G4 structure in this transcript in the plant response to cold. Disruption of this individual RNA G- quadruplex promoted accelerated mRNA decay in the cold of the transcript. This was also correlated with a root growth phenotype in the cold. The paper is very interesting and original and I suggest some comments to improve the paper:

1. The quantification of immunofluorescence signal per μm^2 in Fig. 2a is rather tricky as the spots have different sizes and may hide tendencies as cells have different sizes too. What about foci per cell as cold may have an impact on cell surface? Several normalisations should be considered for these comparisons.
2. The differences of G4 duplexes per loop length are also rather minor in Fig. 2e. Is this biologically relevant? This may need to be better explained.
3. In Fig. 3d the difference between 22 and 4 degrees on mRNA stability are very clear. However, the significant correlation for 3'UTRs RG4 compared to the others may require a better description or even include independent graphs to more clearly highlight the significant differences. Furthermore, when we looked to Extended Fig. 5h the AT4G32020 gene is increased initially after cordycepin addition? Accumulation is increased by cordycepin?
4. The complementation data for the *corg1* mutation with RG4 or mutated CORG1 versions is rather convincing. The phenotype is nevertheless relatively minor, an increase of 20% of root length. Other

characteristics of the cold response can be addressed or even a transcriptome comparison of the two versions to see the global impact on gene expression (e.g. cold-stress markers). There are many cold responses and markers to show that the changes in RG4 structure on the CORG1 have an impact on cold response (and/or the modulation of a kinetics of the cold response?)

5. What is the evidence that these RG4 are “sensing” the cold as suggested? Can in vitro experiments being done to show modulation of RG4 structure by temperature?

Reviewer #1 (Remarks to the Author):

In their manuscript Yang et al. investigate the role of rG4 in plant transcript as a potential marker for cold response. The authors combine a series of bioinformatic analysis with rG4 mapping strategies (SHALiPE) to address the potential biological role of rG4 structure in mRNA stability and plant growth. Using BG4-staining counting cytoplasmic foci (as described in seminal paper by Biffi et al Nat Chem 2013) they quantify a global increase of rG4 formation in cold condition, which is supported by the use of a positive control ligand PDS. By using transcriptome-wide approaches the authors then demonstrate that transcript with rG4 increases mRNA stability. This experiment is particularly elegant and convincing as it is performed on a transcriptome-wide scale and not only of few selected transcript. Furthermore, the authors demonstrate how the presence of such rG4 can be relevant for plant development. Overall, this is a very strong and elegant paper with robust evidence to support the authors conclusion. Therefore I am supportive of publication in Nat Comm.

Response: We thank the Reviewer for their encouraging feedback: “this is a very strong and elegant paper with robust evidence to support the authors’ conclusion”. We are also very grateful for their suggestions in helping us improve this manuscript.

I have only some minor comments for the authors to consider prior publication:

1) The nature of rG4 involved in the mechanism described seems to be multi-tetrad long looped (as mentioned by the authors). It would be interesting to have a UV or CD melting curve for some of these sequences to confirm that the switching temperature between folded/unfolded rG4 is somewhere between 4 and 22 C.

Response: We thank the Reviewer for this very constructive comment. We performed CD to examine RG4 folding under different temperatures by testing two RG4s located on *AT3G20470* and *AT4G29020* (as listed in our TableS4). As shown in Rebuttal Figure 1a-b, for the RG4 on *AT3G20470* (**GGAGGAGGAGG**), the temperature shift from 22 °C to 4 °C induced a positive peak at ~260 nm and a negative peak at ~240 nm, confirming that the folding status of this RG4 was enhanced within this declining temperature range. This result is in good agreement with our *in vivo* SHALiPE-seq results in our main Fig. 2c.

Similarly, the folding status of another RG4 on *AT4G29020* (**GGTCATGGAGTAGGTGG**) was enhanced within the declining temperature range (Rebuttal Figure 1c-d), suggesting

stronger RG4 folding under cold conditions. Thus, both examples showed this low to high switch in the folding state of RG4s in response to high to low temperature changes, i.e. from 22°C to 4°C. We have updated the result for the RG4 on *AT3G20470* in our Extended Data Fig.4 that is associated with Fig. 2c. We have also revised our manuscript correspondingly.

Rebuttal Figure 1. Biophysical characterization for the folding status of two RG4s on *AT3G20470* and *AT4G29020* in response to a decline in temperature change.

(a and c) CD spectra as a function of reducing temperature of the RG4s on *AT3G20470* and *AT4G29020*, respectively. The temperature was reduced from 22°C to 4°C where both the positive peak at ~260 nm and the negative peak at ~240 nm were enhanced. (b and d) The CD signals (ellipticity monitored at 262 nm) as a function of temperature declined from 22°C to 4°C, showing distinct folding status transitions in the folding of the RG4s on *AT3G20470* and *AT4G29020*, respectively.

2) On a similar note it would be good to have some biophysical characterisation of some of the rG4 sequences identified (i.e. CD spectra), just to confirm that those sequences are indeed folding into a G4, although the SHALiPE data are already pretty convincing, but just for completeness.

Response: We thank the Reviewer for this comment. We again performed CD analysis on the RG4 on *AT3G20470* (SHALiPE profile shown in Fig. 2c) under different K^+ concentrations. As shown in Rebuttal Figure 2, the increase of the K^+ concentration induced a positive peak at ~ 260 nm and a negative peak at ~ 240 nm, indicating K^+ -dependent RG4 folding. This result indicates that this RG4 motif is capable of folding into a strong RG4 structure. We have updated the result in revised Extended Data Fig.4a-b. We have also revised our manuscript correspondingly.

Rebuttal Figure 2. Biophysical characterization for the folding status of the RG4 on *AT3G20470* in response to an increase in K^+ concentration.

(a) CD spectra as a function of increasing K^+ concentration. The K^+ ion-induced RG4 folding occurred at a $2.5\mu\text{M}$ RNA in a background of 10mM lithiumcacodylate (LiCac) (pH7.0), as the KCl concentration was increased up to 100mM . Both the positive peak at ~ 260 nm and the negative peak at ~ 240 nm were enhanced with rising K^+ concentration. (b) The CD signal (ellipticity monitored at 262 nm) as a function of increasing K^+ concentration (from 0 to 100mM normal cellular condition) showed the increasingly stronger folding status of this RG4.

3) I found extended data Fig.1 extremely compelling, I wonder whether there is scope for this to be in the main text.

Response: We are pleased that the Reviewer favoured this figure. We agree that it might be helpful to emphasize the specific link between G-nucleotide enrichment and cold climate, and have moved this Rebuttal Figure 3 to become the New Fig.1 b-c in our revised manuscript.

Rebuttal Figure 3 (updated Fig. 1).

Reviewer #2 (Remarks to the Author):

Yang et al analyzed the nucleotide compositions of transcriptomes across plants and conclude that those living in cold climates have G-quadruplex enriched transcriptomes. One thing is to observe that different species living at high or low temperatures have different rG4 propensities. Another entirely different topic is to study G4 formation in a given plant grown at different temperatures. The latter seems relatively trivial: RNA secondary structures, including G-quadruplexes, should be more stable at low temperature. One would expect a similar behavior in human cells for example. Has this been tested?

Response: We are very grateful to the Reviewer for these constructive comments. Our discovery of an association between cold climates and RG4s inspired us to explore the molecular mechanisms underlying this phenomenon. The subsequent temperature-dependent folding of RG4s in regulating mRNA stability, uncovered for the first time, a positive role of RG4 in gene regulation, rather than in inhibiting translation. This contribution of RG4-mediated RNA degradation in response to cold provides a novel in-depth insight into how plants adapt to different environmental conditions.

We agree with the Reviewer that the thermo-capability of RG4s to undergo temperature-dependent folding may have enabled its broad functionality in cold response across the plant kingdom, which may not be limited to plants. Notably, the co-submitted preprint manuscript, entitled “Stress promotes RNA G-quadruplex folding in human cells” has proved that cold also promotes RG4 folding in human cells (Note Figures S1 and S2 in <https://doi.org/10.1101/2022.03.03.482884>). Therefore, this research supports our premise that RG4 folding is generally enhanced in response to cold.

1) Fig 1a: The fact that both G and C are enriched in cold-living plants is not necessarily a proof that G-quadruplexes are actually involved (one would expect only the fraction of G to increase – what is seen here is more like an increase in GC content).

Response: We thank the Reviewer for this insightful comment. From a DNA sequence aspect, G and C contents have been studied together as “GC content”. Nevertheless, G content is not necessarily tightly linked with C content in RNA. In our study, the correlation between climatic parameters and G content did not tightly synchronize with C content. We found 12 negative correlations between climatic parameters and G content. But we found only 4 negative correlations and 2 positive correlations between climatic parameters and C content (original

Extended Data Fig 1 and revised main Fig 1). Therefore, we concluded that G content exhibits a climatic signature of plant adaptation. This discovery further inspired us to explore the functional importance of RG4 in plant response to the cold.

2) Fig 1b: the warm/cold data is strongly skewed by bryophytes: all of them live in cold environments, and their G4 density is clearly higher than in other plant species.

Response: We thank the Reviewer for this comment. Bryophytes evolved as an important plant clade. The high frequency of RG4s in the Bryophyte transcriptome is likely to facilitate its survival in cold environments. We also calculated the mean RG4 density and the mean annual mean temperature for each clade (Rebuttal Figure 4). As RG4 density increased, the annual mean temperatures generally reduced across clades with a negative correlation of -0.61.

Rebuttal Figure 4. The RG4 density and the annual mean temperature across different plant clades.

(a) Box plot showing the RG4 density for different clades of plants that were ranked from low RG4 density to high RG4 density.

(b) Box plot showing the annual mean temperature of their habitats for different clades of plants that were ranked from low RG4 density to high RG4 density as indicated in Rebuttal Figure 4a.

3) Some of the final statements need to be tuned-down, such as “G-quadruplex structure

serving as a molecular marker shaping plant environmental adaptation” or “RNA G-quadruplexes embedded across the plant transcriptome globally facilitate stabilizing RNA molecules in response to cold”.

Response: We thank the reviewer for this comment, and confirm that in our updated revision we have tuned down the statements, thus: “G-quadruplex structure may serve as a molecular marker in shaping plant environmental adaptation”; and “RNA G-quadruplexes embedded across the plant transcriptome might play an important role in stabilizing RNA molecules in response to cold.”

4) Detail: “RNA G-quadruplex (RG4), involving both Hoogsteen and Watson-Crick base pairs” is a bit misleading – most quadruplexes involved only Hoogsteen type base pairing. Overall, while potentially interesting, this manuscript is more appropriate for a specialized plant journal.

Response: We thank the reviewer for this comment. We have modified the phrase to: “RNA G-quadruplex (RG4), involving the base pairs on both Hoogsteen and Watson-Crick faces”. In addition, we have added the figures illustrating the base pairs for RG4 formation in Extended Data Fig. 1a, as shown in Rebuttal Figure 5.

Rebuttal Figure 5. Schematic illustration of RNA G-quadruplex structure.

We would like to emphasize that RNA G-quadruplex is one of the important RNA structure motifs in organisms. In eukaryotes, the existence of RG4 was first found in plants (Yang et al., 2020), providing the research hypothesis that RG4 might also exist and function in other eukaryotic organisms, such as mammalian cells. While we further explored the functional importance of RG4 in plants, several groups were simultaneously researching whether RG4 has a functional role in animals. Indeed, the co-submitted manuscript (“**Stress promotes RNA**

G-quadruplex folding in human cells”) from Prof. Ivanov (Harvard) and Prof. Junjie Guo (Yale) has revealed that hundreds of RNA G-quadruplexes in human cells are rapidly induced by physiological stresses and serve as an mRNA stabilizer in facilitating the survival of human under stress. Our collective research in plants (Ding and Zhang) and human (Ivanov and Guo) demonstrate exciting coherence into the functional importance that RNA G-quadruplex has in stress response. Therefore, we believe that our distinct and complementary discoveries will together significantly impact a wide range of specialists within the scientific community. As such, we believe that our manuscript perfectly suits the broad audience of *Nature Communications*.

Reviewer #3 (Remarks to the Author):

The authors analysed nucleotide compositions of transcriptomes across 1000 plants and compared them with their corresponding climate to observe an interesting correlation between G-rich transcriptomes and G-quadruplex structures and cold acclimated plants. They use immunofluorescence and a variety of genomic approaches (including SHALiPEseq, cordycepin treatments for mRNA seq) to assess the impact of G4 quartets on translation efficiency and mRNA stability at different temperatures and find a strong correlation with cold responses. Then, they identified a specific target CORG1, a cold regulated G4-quartet responsive transcript, to address the relevance of the G4 structure in this transcript in the plant response to cold. Disruption of this individual RNA G- quadruplex promoted accelerated mRNA decay in the cold of the transcript. This was also correlated with a root growth phenotype in the cold. The paper is very interesting and original and I suggest some comments to improve the paper:

Response: We are very grateful to the Reviewer for their appreciation that “The paper is very interesting and original and I suggest some comments to improve the paper”. We thank the Reviewer for their very constructive comments to improve the manuscript.

1) The quantification of immunofluorescence signal per μm^2 in Fig. 2a is rather tricky as the spots have different sizes and may hide tendencies as cells have different sizes too. What about foci per cell as cold may have an impact on cell surface? Several normalisations should be considered for these comparisons.

Response: We thank the Reviewer for this useful comment, and have reanalysed our data and calculated foci per cell under different conditions, accordingly. As shown in Rebuttal Figure 6,

we found that the BG4 signal under cold conditions is much stronger than that under control conditions, confirming that cold promotes RG4 folding.

Rebuttal Figure 6. Reanalysis of the immunofluorescence detection of G-quadruplex (RG4) with a BG4 antibody in *Arabidopsis* under different thermal conditions. *Arabidopsis* seedlings were grown at 22°C (Control) and were: i) treated at 4°C for 2 hours (Cold); or ii) returned to 22°C for 2 hours after cold treatment (Recovery); or iii) treated with 10 µM pyridostatin (PDS); or iv) treated with RNase cocktail (RNase). The foci per cell were calculated. More than 80 cells from 3 individual seedlings for each condition were subjected to statistical analysis and were significance tested by Student's t-test in comparison to control.

2) *The differences of G4 duplexes per loop length are also rather minor in Fig. 2e. Is this biologically relevant? This may need to be better explained.*

Response: We are grateful to the Reviewer for this comment. We further performed gene ontology analysis of RG4s genes with different loop lengths. Interestingly, we found that gene functions are quite distinct among those genes containing RG4s with different loop lengths (Rebuttal Figure 7). For instance, RG4 genes with a loop length of 2nt were associated with functions related to the response to plant hormones, such as salicylic acid (SA), and abscisic acid (ABA). The genes containing RG4s with a loop length of 4nt were enriched in the functions relating to translation, suggesting these RG4s may contribute to the regulation of translation in response to cold. The genes containing RG4s with longer loops (6nt, 8nt, 10nt) were mostly enriched in the functions related to abiotic stress responses such as temperature, light, and water. This suggests that the genes containing RG4s with long loops may participate in responses to different environmental conditions, which may be of great interest for future investigations.

Rebuttal Figure 7. Detailed gene ontology analysis of genes containing cold-responsive RG4s with different loop lengths.

3) In Fig. 3d the difference between 22 and 4 degrees on mRNA stability are very clear. However, the significant correlation for 3'UTRs RG4 compared to the others may require a better description or even include independent graphs to more clearly highlight the significant differences. Furthermore, when we looked to Extended Fig. 5h the AT4G32020 gene is increased initially after cordycepin addition? Accumulation is increased by cordycepin?

Response: We thank the Reviewer for this comment. We emphasized the differences between 3'UTR and other regions in our original Fig. 3f which illustrates a significantly greater difference in decay rate between 4°C and 22°C on transcripts with cold-responsive RG4s in 3'UTR compared to that of transcripts without RG4s or with RG4s in 5'UTR or CDS regions.

For the decay profile of *AT4G32020* in Extended Fig.5h, the initial increase of mRNA abundance after cordycepin addition is most likely caused by the high stability of this transcript at 4°C. Due to its much higher stability in comparison with the vast majority of transcripts, the RNA level of *AT4G32020* increased, in contrast to the total pool of RNAs in the early stage of cordycepin treatment. This phenomenon was observed for stable transcripts in other studies, such as Sorenson et al., 2018, Proc Natl Acad Sci (PMID: 29386391).

4) The complementation data for the *corg1* mutation with *RG4* or mutated *CORG1* versions is rather convincing. The phenotype is nevertheless relatively minor, an increase of 20% of root length. Other characteristics of the cold response can be addressed or even a transcriptome comparison of the two versions to see the global impact on gene expression (e.g. cold-stress markers). There are many cold responses and markers to show that the changes in *RG4* structure on the *CORG1* have an impact on cold response (and/or the modulation of a kinetics of the cold response?)

Response: We thank the Reviewer for this insightful comment. An *RG4* mutation in 3'UTR causing an increase of 20% of root length after 4 weeks of cold treatment is quite strong since there are quite a few of genes in *Arabidopsis* that regulate the cold-responsive root growth (Guo et al., *Journal of Integrative Plant Biology*, 2018). We also measured the plant size where we found that *mutRG4-CORG1* plants were overall bigger compared to *wtRG4-CORG1* plants after cold treatment, further supporting the functional importance of this *RG4* (Rebuttal Figure 8 and Extended Data Fig. 6f).

Rebuttal Figure 8. RNA G-quadruplex regulates plant cold-responsive growth.

Phenotypes of plants of different genotypes grown at 22°C or 4°C. *wtRG4* or *mutRG4* denotes *corg1-1* mutant complemented with genomic DNA of *CORG1* carrying *wtRG4* or *mutRG4*

respectively, as illustrated in Fig. 4A. Comparison was performed on plant growth at 22°C for 1 week, or at 4°C for 4 weeks.

Following the reviewer’s suggestion, we measured the expression levels of marker genes for *Arabidopsis* cold response in wtRG4 plants and mutRG4 plants with cold treatment (4°C), including key transcription factors CBF1, CBF2, CBF3 and their target gene COR15A. As shown in Rebuttal Figure. 9, mRNA abundances of marker genes upon cold treatment are significantly lower in mutRG4 plants compared to those of wtRG4 plants. The temperature-dependent changes in expression levels of these cold marker genes further support our finding that the changes of RG4 structure on CORG1 have a general impact on plant cold response. The new results have been added in Extended Data Fig. 7.

Rebuttal Figure. 9. RNA G-quadruplex contributes to cold-responsive gene expression.

Relative mRNA abundances of marker genes *CBF1*, *CBF2*, *CBF3* and *COR15A* upon cold treatment (0, 3, 24 hours) in complemented *corg1-1* mutant using genomic DNA sequence of *CORG1* carrying wtRG4 or mutRG4 in 3’UTR, respectively. Error bar indicates SE of 4 biological replicates.

5) What is the evidence that these RG4 are “sensing” the cold as suggested? Can *in vitro* experiments being done to show modulation of RG4 structure by temperature?

Response: We thank the Reviewer for this comment, which was also suggested by Reviewer #1 (Comment 1). Briefly, we performed CD to examine RG4 folding under different temperatures by testing two RG4s located on *AT3G20470* and *AT4G29020* listed in our tableS4. We found that cold promotes RG4 folding, suggesting that these identified RG4s folded in

response to temperature. This cold-induced folding property of these identified RG4s is likely to contribute to plant sensing of the cold.

REVIEWERS' COMMENTS

Reviewer #1 (Remarks to the Author):

The authors have successfully addressed my comments and I am now supportive of publication of this manuscript in Nat Comm

Reviewer #1 comments on Reviewer #2 general comment and previous comment 2 (Remarks to the Author):

The way I interpreted this is that the authors were implying this was a correlation type study/analysis, but I think that asking them perhaps to spell it out clearly might not hurt to make sure there is no misleading message sent there.

Overall I agree that the correlation might not be straightforward but there are also other factors contributing to temperature dependent organisms beyond rG4 so I would not expect a clear cut correlation. Possibly the best way forward is to ask the authors to caveat this and discuss it more if they want to keep this, so that there is a more fair projection and illustration of the data and hypothesis presented?

Reviewer #3 (Remarks to the Author):

The authors have addressed adequately all my previous comments in this revised version. Furthermore, the addition of in vitro data for the temperature dependent conformational changes in this new version makes the results even more interesting. I think this paper is a very nice piece of work.

REVIEWERS' COMMENTS

Reviewer #1 (Remarks to the Author):

The authors have successfully addressed my comments and I am now supportive of publication of this manuscript in Nat Comm

Response: We thank the Reviewer in supporting our manuscript. We appreciate the time and effort the Reviewer has spared.

Reviewer #1 comments on Reviewer #2 general comment and previous comment 2 (Remarks to the Author):

The way I interpreted this is that the authors were implying this was a correlation type study/analysis, but I think that asking them perhaps to spell it out clearly might not hurt to make sure there is no misleading message sent there.

Overall I agree that the correlation might not be straightforward but there are also other factors contributing to temperature dependent organisms beyond rG4 so I would not expect a clear cut correlation. Possibly the best way forward is to ask the authors to caveat this and discuss it more if they want to keep this, so that there is a more fair projection and illustration of the data and hypothesis presented?

Response: We thank the Reviewer's suggestion. We have now added a sentence in the Discussion session, as "*Correlations between RG4 frequency and temperature bioclimatic variables are not very high (Fig.1e), suggesting other factors besides RG4 may also contribute to plant thermal adaptation*".

Reviewer #3 (Remarks to the Author):

The authors have addressed adequately all my previous comments in this revised version. Furthermore, the addition of in vitro data for the temperature dependent conformational changes in this new version makes the results even more interesting. I think this paper is a very nice piece of work.

Response: We thank the Reviewer for their positive comments.